# The Antiviral Effects of Heat-Killed *Lactococcus lactis* Strain Plasma Against Dengue, Chikungunya, and Zika Viruses in Humans by Upregulating the IFN-α Signaling Pathway

**DOI:** 10.3390/microorganisms12112304

**Published:** 2024-11-13

**Authors:** Zhao Xuan Low, Osamu Kanauchi, Vunjia Tiong, Norhidayu Sahimin, Rafidah Lani, Ryohei Tsuji, Sazaly AbuBakar, Pouya Hassandarvish

**Affiliations:** 1Tropical Infectious Disease Research and Education Centre (TIDREC), Universiti Malaya, Kuala Lumpur 50603, Malaysia; lowzhaoxuan@um.edu.my (Z.X.L.); kanauchio@wonderfarmonline.com (O.K.); ayusahimin@um.edu.my (N.S.); sazaly@um.edu.my (S.A.); 2Institute of Health Sciences, Kirin Holdings Co., Ltd., 2-26-1, Muraoka-Higashi, Fujisawa 251-8555, Kanagawa, Japan; ryohei_tsuji@kirin.co.jp; 3Department of Medical Microbiology, Faculty of Medicine, Universiti Malaya, Kuala Lumpur 50603, Malaysia; rafidahl@um.edu.my

**Keywords:** LC-Plasma, antiviral, arboviruses, interferon, probiotics

## Abstract

The growing risk of contracting viral infections due to high-density populations and ecological disruptions, such as climate change and increased population mobility, has highlighted the necessity for effective antiviral treatment and preventive measures against Dengue virus (DENV), Chikungunya virus (CHIKV), and Zika virus (ZIKV). Recently, there has been increasing attention on the use of probiotics as a potential antiviral option to reduce virus infections. The present study aimed to assess the immunomodulatory effects of heat-killed *Lactococcus lactis* strain plasma (LC-Plasma) on peripheral blood mononuclear cells (PBMCs) and its subsequent antiviral response against DENV, CHIKV, and ZIKV. To evaluate the immunomodulatory effects of LC-Plasma on PBMCs isolated from healthy individuals, PBMCs were cultured at a density of 2 × 10^5^ cells/well and stimulated with 10 µg/mL of LC-Plasma. LC-plasma-stimulated PBMCs demonstrated elevated interferon-alpha (IFN-α) production and cluster of differentiation 86 (CD86) and human leukocyte antigen-DR isotype (HLA-DR) upregulation, potentially linked to plasmacytoid dendritic cell (pDC) activation. The replication of DENV, CHIKV, and ZIKV was dose-dependently inhibited when Huh-7 cells were stimulated with LC-Plasma-stimulated PBMC supernatant (LCP Sup). IFN-stimulated gene (ISG) expression, including IFN-stimulated gene 15 (ISG15), IFN-stimulated exonuclease gene 20 (ISG20), IFN-induced transmembrane protein 1 (IFITM-1), myxovirus resistance protein A (MxA), and radical S-adenosyl methionine domain-containing protein 2 (RSAD2), was significantly upregulated in LCP Sup-stimulated Huh-7 cells. Findings from this study indicate that LC-Plasma has the potential to induce IFN-α production, leading to an enhancement in the expression of ISGs and contributing to a broad-spectrum antiviral response. Thus, LC-Plasma may serve as a rational adjunctive treatment to ameliorate viral diseases, warranting future clinical trials.

## 1. Introduction

Arboviruses are viruses transmitted by arthropods, such as mosquitoes. Infection with these viruses contributes substantially to morbidity and mortality in tropical and subtropical regions due to the abundance of mosquitoes in these areas [1]. In recent years, an upward trend has emerged as certain arboviruses, namely, DENV, CHIKV, and ZIKV, have expanded their geographic reach [2]. Tropical viral infections, like Dengue, Chikungunya, and Zika, have become major global health concerns. Currently, 2.5 billion people are at risk of Dengue, with 390 million infections yearly, leading to 500,000 hospitalizations and 25,000 deaths [3]. In 2023, Chikungunya caused recurrent outbreaks in South Asia, affecting around 500,000 people and resulting in over 400 deaths [4,5]. Since spreading from Africa to Brazil in 2015, Zika has infected over one million people and is now present in 84 countries [6]. These viruses pose a growing threat worldwide, particularly in temperate regions.

Numerous viral infectious diseases lack targeted therapeutics, relying exclusively on supportive clinical intervention. Based on the World Health Organization, there are no specific antivirals available for DENV, CHIKV, and ZIKV infections or effective vaccines for Zika fevers [7,8]. Although newly developed vaccines for DENV and CHIKV are available, they still possess limitations. For example, the long-term effectiveness of the QDENGA TAK-003 (Takeda, Japan) and IXCHIQ (Valneva, France) vaccines against DENV and CHIKV, respectively, remains uncertain. Additionally, the use of DENGVAXIA (Sanofi Pasteur, France) for DENV is restricted due to safety and efficacy concerns for primary Dengue patients [9]. Consequently, there is an urgent need to explore alternatives that complement existing treatments, potentially incorporating nutraceutical interventions to combat these tropical viruses. Recently, there has been increasing attention on the use of probiotics as a potential antiviral option to reduce the severity of symptoms and healthcare burden related to virus infections [10]. Probiotics are living microorganisms, which confer health benefits to the host when ingested in adequate amounts [11].

*Lactococcus lactis* strain plasma (LC-Plasma), also known as *Lactococcus lactis* subsp. *lactis* JCM5805, can directly stimulate pDCs to produce Type I and Type III IFNs [12]. LC-Plasma stimulates pDCs to produce Type I IFN via TLR9 and, subsequently, induces the upregulation of ISGs, such as ISG15, Mx1, Oasl2, and viperin in HepG2 cells and intestinal epithelial cells (IECs), thus inhibiting the replication of DENV and rotavirus, respectively [13,14]. Besides, LC-Plasma also activated NK cells via dendritic cells, thus increasing the production of IFNγ and granzyme in NK cells, subsequently enhancing the cytotoxicity activity of NK cells [15]. Moreover, oral administration of LC-Plasma could restore and maintain the activity of pDCs in heat-stressed mice [16], potentially protecting against tropical infectious viruses, which are more likely to spread in hot seasons or regions due to heightened thermal stress. The effects of IFN-α induction by LC-Plasma stimulation are well established, but the broad-spectrum antiviral activity against major tropical arboviruses has not been substantiated. This study assessed the antiviral effects of LC-Plasma against DENV, CHIKV, and ZIKV using an in vitro infection model with Huh-7 cells and human PBMCs as a source of LC-Plasma-induced humoral factors. We hypothesized that LC-Plasma can stimulate PMBCs to upregulate the production of IFN-α and, subsequently, induce an antiviral state in vitro, thus reducing the viral infectivity of DENV, CHIKV, and ZIKV.

In combating viral infections, one of the desired immunomodulatory effects of probiotics is the stimulation of IFNs [17]. IFNs are cytokines produced by immune cells for protection against viruses [18]. IFNs are classified into three major groups, Type I, Type II, and Type III. Type I IFNs, such as IFN-α and IFNβ, are the major effector cytokines of the host immune response against viral infections [19]. IFN-α is mainly produced from pDCs upon stimulation by viral nucleic acids via TLR7, 8, and 9 [20,21]. Type I IFN can induce the expression of IFN-stimulated genes (ISGs) via Type I IFN receptors (IFNAR), which are highly expressed on almost all cell surfaces [22]. ISGs have diverse physiological activities against virus infection by targeting various viral life cycle stages [23]. For example, ISG15, a ubiquitin-like IFN-stimulated protein, can conjugate viral proteins, thus inhibiting viral release [24]. Besides IFNs, immune cells also produce other cytokines, such as interleukins (ILs), which play an indirect immunoregulatory function in the immune response to viral infection [25]. However, excessive production of these pro-inflammatory cytokines (overwhelming immune reaction) leads to cytokine storms, causing organ damage and worsening disease progression [26]. Thus, immune modulation in balancing the pro- and anti-inflammatory states is crucial for combating viral infections.

## 2. Materials and Methods

### 2.1. The Study Design

A total of 25 healthy Malaysian volunteers aged between 18 and 35 were recruited for blood sampling. Subjects enrolled in this study were healthy adults with normal heart rate (60–100 beats per minute), body temperature (36.1 °C to 37.2 °C), and blood pressure (120/80 mmHg), as verified by a certified medical physician. Informed consent was obtained from all participants. Parameters obtained from a previous study [27] were used to determine the sample size (effect size: 0.8–0.9, type 1 error: 0.05, and power: 0.8–0.9). The Medical Research Ethics Committee of Universiti Malaya Medical Centre approved this study in 6 May 2022 (UMMC MREC No. 202256-11216).

### 2.2. PBMC Isolation, Preservation, and Culture

PBMCs were isolated from whole blood using a BD Vacutainer CPT glass molecular diagnostics tube (BD, Franklin Lakes, NJ, USA), according to the manufacturer’s protocol. Briefly, whole blood was collected using standard venipuncture and centrifuge within two hours at 1600 RCF (Relative Centrifuge Force) for 30 min. PBMCs were washed twice with phosphate-buffered saline (PBS), and aliquots of the PBMC pellet were resuspended in Cellbanker2 (ZENOGEN, Koriyama, Japan) and cryopreserved at −80 °C. Upon thawing, PBMCs were cultured in Roswell Park Memorial Institute (RPMI)-1640 media (Corning, Corning, NY, USA) supplemented with 1% sodium pyruvate (1 mM) (Gibco, Waltham, MA, USA), 0.25% 4-(2-hydroxyethyl)-1-piperazineethanesulfonic acid (HEPES) (2.5 mM) (Gibco, USA), 1% penicillin/streptomycin (50 units/mL) (Gibco, USA), 10% fetal bovine serum (FBS) (Gibco, USA), and 1% nonessential amino acid (NEAA) (Gibco, USA) [27]. PBMC viability was evaluated using trypan blue (Bio-rad, Hercules, CA, USA) staining, and the percentage of viability was calculated using a TC20 automated cell counter (Bio-rad, USA). Upon thawing, PBMC viability was 80% and above.

### 2.3. Cells

Vero cells (African Green Monkey kidney) (ECACC strain) were cultured and maintained in high glucose (4 g/L) Dulbecco’s Modified Eagle Medium (DMEM) (Corning, USA). In contrast, Huh-7 cells (JCRB Cell Bank JCRB0403) were cultured and maintained in low glucose (1 g/L) DMEM (Corning, USA). For cell growth, all media were supplemented with 10% heat-inactivated FBS (Gibco, USA). At the time of virus inoculation and antiviral assays, the concentration of FBS was reduced from 10% to 2%. All cells were incubated at 37 °C in a 5% CO_2_ humidified atmosphere, maintained in a T75 cell culture flask, and subcultured once confluent (3–4 days once) [28,29].

### 2.4. Viruses

DENV-2 (New Guinea C (NGC) strain, accession number: ATCC VR-1584), CHIKV (ECSA genotype, MY/065/08/ accession number: FN295485), and ZIKV (strain P6740, obtained from the University of Texas Medical Branch) were propagated in Vero cells. DENV was titrated using foci assay, while CHIKV and ZIKV were titrated using plaque assay.

### 2.5. Infectious Virus Particle Titration (Plaque and Foci Assay)

Due to the delayed cytopathic effect exhibited by DENV, its viral titer was quantified using a foci assay, while CHIKV and ZIKV were quantified using a plaque assay. Vero cells were seeded overnight at a density of 1 × 10^5^ cells/well in 24-well plates. Monolayer confluent Vero cells were infected with ten-fold serial diluted DENV, CHIKV, and ZIKV for one hour at 37 °C for preadsorption. Thereafter, the inoculum was removed, and the cells were overlaid with immobilizing media comprising 0.9% high-viscosity carboxymethyl cellulose (CMC) (Sigma Aldrich, Saint Louis, MO, USA) supplemented with 2% FBS. The DENV, CHIKV, and ZIKV-infected Vero cells were further incubated for 4, 2, and 5 days, respectively, at 37 °C to allow the formation of countable foci or plaque. After incubation, immobilizing media was removed and fixed with 4% paraformaldehyde (MP Biomedicals, Irvine, CA, USA). After fixation, for DENV foci assay, the cells were permeabilized using 100-fold diluted Triton X-100 (Sigma-Aldrich) at room temperature for 15 min. After washing, the primary antibody, composed of serum from a DENV-recovered patient and diluted (1:500) in 3% skim milk (Thermo Fisher, Oxford, UK), was added and incubated for one hour at 37 °C. After washing, horseradish peroxidase (HRP)-conjugated rabbit antihuman IgG (H+L) secondary antibody (Invitrogen, Waltham, MA, USA) was added at a ratio of 1:1000 and further incubated for one hour at 37 °C. Lastly, TrueBlue peroxidase substrate (Seracare, Milford, MA, USA) was added to each well in the dark [29]. Foci in each well were counted, and viral titer was determined in foci forming unit per microliter (FFU/mL). For the visualization of the plaque assay for CHIKV and ZIKV, the infected cells were stained with 0.5% crystal violet at 4 °C for one hour. The plaques in each well were counted, and viral titer was determined in the plaque-forming unit per microliter (PFU/mL) [30].

### 2.6. Preparation of LC-Plasma Suspension

*Lactococcus lactis* strain plasma (LC-Plasma), also known as *Lactococcus lactis* subsp. *lactis* JCM5805, was heat-killed, lyophilized, and kept at 4 °C, away from light exposure. The lyophilized LC-Plasma was first reconstituted in RPMI 1640 media (Corning, USA) and dispersed using an ultrasonic homogenizer by Omni-Ruptor 4000 (OMNI international, Kennesaw, GA, USA) for two minutes at 40 watts on ice. LC-Plasma was then diluted to a final concentration of 10 μg/mL during PBMC stimulation. Fresh LC-Plasma was prepared for each round of PBMC stimulation.

### 2.7. Supernatant of PBMCs

PBMCs were thawed and washed with 10 mL of PBS and culture at a density of 2 × 10^5^ cells/mL in each well of 96-well plates. The PBMCs were then stimulated with LC-Plasma at a concentration of 10 μg/mL for 24 h [1,3,5,6]. As a positive control for IFN-α, 1 μM of CpG ODN 2216 (Invivogen, San Diego, CA, USA), a representative TLR9 ligand, was used to stimulate PBMCs for 24 h. Thereafter, the PBMC supernatants were collected by centrifugation at 300 RCF for 5 min, aliquoted, and stored at −80 °C. Freeze-thawing of these PBMC supernatants was avoided [14].

### 2.8. Enzyme-Linked Immunosorbent Assay (ELISA) for IFN-α Quantification

The concentrations of IFN-α in PBMC supernatants were measured using the Human IFN-α ELISA Kit (Elabscience, Wuhan, China, E-EL-H6125), according to the manufacturer’s protocols. Optical density (OD) value was measured at 450 nm using a microplate reader, infinite 200 Pro (Tecan, Männedorf, Switzerland). Duplicate OD readings for each standard and sample were averaged and then subtracted with OD reading from blank. The IFN-α concentration was determined by interpolating from a four-parameter logistic curve, constructed using GraphPad Prism 9.0 based on the OD values obtained from known concentration standards. The actual IFN-α concentration in the PBMC supernatants was calculated by multiplying it with the dilution factor.

### 2.9. Immunophenotyping via Flow Cytometry

After 24 h of treatment with LC-Plasma and CpG ODN 2216, PBMCs were analyzed through immunophenotyping to assess the upregulation of activation markers on gated pDCs. To address immunological variability, PBMCs from individual donors were stimulated separately and then pooled before immunophenotyping. The stimulated PBMCs from five participants were randomly combined to form five distinct pools, which were used as biological replicates (N = 5) during the immunophenotyping process. Pooling was necessary to acquire sufficient number of live pDCs (at least 150 events/pooled sample) during flow cytometry, as pDCs account for only a small fraction of the total PBMC population (0.2–0.8%). The expression of activation markers HLA-DR and CD86 in pDCs from PBMCs was assessed by flow cytometry following the previous publication by Komano et al. [31]. Briefly, LC-Plasma-stimulated PBMCs were washed and stained with antihuman CD304/B515, CD86/PE, HLA-DR/PE-CyTM, and CD123/APC antibodies (BD, USA) for 30 min at 4 °C and 7AAD cell viability dye (BD, USA) for five minutes at 4 °C. After incubation, cells were resuspended in a staining buffer (BD, USA) and further analyzed with a BD FACScanto II flow cytometer (BD, USA). pDC is defined as CD123+ and CD304+ cells. Data were analyzed using FACSDiva software version 6.1.2 (BD, USA). An unstained sample was prepared to adjust for cellular autofluorescence to eliminate the background signal.

### 2.10. Antiviral Assay

Huh-7 cells were pre-seeded in 24-well plates at 1 × 10^5^ cells/well overnight. Monolayer of Huh-7 cells was then stimulated with diluted PBMC supernatant or 100 units of recombinant universal Type I IFN-α (PBL assay science, USA) for 24 h before virus infection. After stimulation, Huh-7 cells were infected with infectious virus particles at a multiplicity of infection (MOI) of 0.1 for one hour at 37 °C. The viral progeny in the supernatant was harvested 48 h postinfection, aliquoted, and stored at −80 °C. The viral load was then quantified using qRT-PCR and foci or plaque assay. To neutralize the activity of the Type 1 IFN signaling pathway, specific blocking antibodies against IFNAR2 (clone MMHAR-2, PBL Assay Science, Piscataway, NJ, USA) were added at a concentration of 3 µg/mL to the culture media. The IFN receptors on Huh-7 cells were blocked for one hour before being stimulated with PBMC supernatant. Simultaneously, mouse IgG2a (Invitrogen, USA) was used as isotype control, ensuring that the abrogation of antiviral effects is due to IFNAR2-specific blocking rather than nonspecific interactions [14].

### 2.11. Quantitative RT-PCR of DENV, CHIKV, and ZIKV

Viral RNA was extracted using the QIAampR Viral RNA extraction kit (QIAGEN, Hilden, Germany, QIAG-52904), following manufacturing protocol. The extracted viral RNA was reverse-transcribed and amplified using a Step-OnePlus Real-Time PCR System with SensiFast SYBER Hi-ROX one-step kit (Bioline, London, UK, BIO-73005) following the manufacturer’s protocol. The primers were specific for DENV capsid gene (forward: CAATATGCTGAAACGCGAGAGAAA, reverse: AAGACATTGATGGCTTTTGA), CHIKV E1 gene (forward: TCGACGCGCCCTCTTTAA, reverse: ATCGAATGCACCGCACACT), and ZIKV E gene (forward: CGAGGACAGGCCTTGACTTT, reverse: ACTCCTTGTGCACCAACCAA).

### 2.12. Gene Expression Study in Huh-7 Cells

A 10-fold diluted PBMC supernatant was used to stimulate Huh-7 cells for 24 h. Afterward, the whole-cell RNA of stimulated Huh-7 cells was extracted using RNeasy Kit (Qiagen, Germany, QIAG-74106), and cDNA was synthesized using the iScript cDNA synthesis kit (BioRad, USA, 1708891), according to the manufacturer’s protocol. ISG expression of stimulated Huh-7 cells was performed using the qRT-PCR method, using SYBR green-based Luna^®^ Universal qRT-PCR Master Mix (NEB, Ipswich, MA, USA, M3003L) with the QuantStudio™ 5 Real-Time PCR System. Relative mRNA expression of the ISGs was calculated using 2^−ΔΔCt^ method [32]. The β-actin (ACTB) gene was used as a reference gene for normalization. The primers used to detect the gene of interest, including RyDEN, IFITM-1, OAS-1, ISG15, ISG20, RSAD2, and Mx-A, were adapted from a previous publication [14].

### 2.13. Statistical Analysis

All values are expressed as median ± SD for 5 biological replicates (N = 5). Data were analyzed by nonparametric Kruskal–Wallis and Steel tests. All statistical analyses were performed using the EZR software version 1.68 program [33].

## 3. Results

### 3.1. LC-Plasma Increases the IFN-α Production of PBMCs

To evaluate the immunomodulatory effects of LC-Plasma, PBMCs were cultured at a density of 2 × 10^5^ cells/well and stimulated with 10 µg/mL of LC-Plasma and 1 µM of CpG ODN 2216 (Invivogen, USA) for 24 h. The latter served as a positive control for IFN-α production. The treatment of PBMCs with CpG ODN 2216 (1 µM) and LC-Plasma (10 µg/mL) resulted in a significant increase in IFN-α production (*p* < 0.05) by 11.44 ± 9.73-fold and 186.20 ± 542.20-fold, respectively (Figure 1).

### 3.2. LC-Plasma Increases the Expression of Cell Surface Activation Markers in the pDCs Population in PBMCs

The upregulation of HLA-DR and CD86 on pDCs from CpG ODN 2216 and LC-Plasma stimulated PBMCs was investigated using an immunophenotyping assay with flow cytometry. The median fluorescence intensity (MFI) levels of HLA-DR in pDCs were increased by 1.50 ± 0.69-fold and 1.39 ± 0.75-fold after CpG ODN 2216 and LC-Plasma treatment, respectively (Figure 2, left panel). The MFI levels of CD86 in pDCs were increased by 4.24 ± 14.31-fold and 2.34 ± 4.76-fold after CpG ODN 2216 and LC-Plasma treatment, respectively (Figure 2, right panel). There was a statistically significant (nonparametric Kruskal–Wallis and Steel test; *p* < 0.05) increase in the expression level of HLA-DR due to LC-Plasma treatment. However, the increase in CD86 levels in pDCs following LC-Plasma treatment did not reach statistical significance, despite exhibiting a higher fold change (2.34-fold) compared to HLA-DR (1.39-fold).

### 3.3. Supernatant Derived from LC-Plasma-Stimulated PBMCs Contains Humoral Factors That Inhibit the Replication of DENV, CHIKV, and ZIKV

A 10-fold diluted PBMC supernatants obtained after stimulation with LC-Plasma (LCP Sup 1:10) reduced infectious virus titers for DENV, CHIKV, and ZIKV by 2.15 ± 0.67, 3.44 ± 2.36, and 2.51 ± 0.64 log, respectively (Figure 3 and Appendix A). Statistical analysis using the nonparametric Steel test suggested a statistically significant reduction (*p* < 0.05) in infectious viral particles for CHIKV and ZIKV, while the anti-DENV effect of LCP Sup 1:10 showed a trend towards reduction (*p* = 0.052). Additionally, LCP Sup 1:10 reduced DENV, CHIKV, and ZIKV viral RNA copy numbers by 1.50 ± 0.71, 2.76 ± 1.60, and 1.53 ± 0.77 log, respectively (Figure 3 and Appendix A). The reduction in the viral RNA copy number of CHIKV was statistically significant, but the anti-DENV and anti-ZIKV effects of LCP Sup 1:10 showed a tendency toward reduction (*p* = 0.052, *p* = 0.08, respectively). The dose-dependent antiviral effects against DENV, CHIKV, and ZIKV were observed with the 10-fold serial dilution of LC-Plasma-stimulated PBMC supernatant (Figure 3 and Appendix A).

The PBMC supernatants obtained following stimulation with the positive control CpG ODN 2216 (CpG Sup 1:10) significantly (*p* < 0.05) reduced the infectious virus titers of DENV, CHIKV, and ZIKV by 5.63 ± 2.02, 6.48 ± 1.53, and 3.95 ± 2.00 log, respectively. Moreover, the viral RNA copy numbers for DENV, CHIKV, and ZIKV were significantly reduced by 3.37 ± 0.88, 4.97 ± 0.32, and 4.23 ± 0.74 log, respectively (Figure 3 and Appendix A). The supernatant of unstimulated PBMCs (Neg Sup 1:10) did not affect DENV, CHIIKV, and ZIKV replication, as evidenced by results from infectious virus assay (plaque/foci) and qRT-PCR. For the positive control, 100 units of recombinant IFN-α (PBL assay science, USA) were used instead of the PBMC supernatant. The infectious virus titers of DENV, CHIKV, and ZIKV were also significantly reduced (*p* < 0.05) by 4.86 ± 0.77, 6.11 ± 0.97, and 5.15 ± 1.27 log and likewise for the viral RNA copy number (*p* < 0.05) by 2.95 ± 0.55, 4.49 ± 0.53, and 3.34 ± 0.44 log, respectively (Figure 3 and Appendix A).

### 3.4. Supernatant Derived from LC-Plasma-Stimulated PBMCs Upregulates the Expression of ISGs of Huh-7 Cells

Huh-7 was stimulated with recombinant IFN-α (100 Units), CpG Sup 1:10, LCP Sup 1:10, or Neg Sup 1:10 for 24 h, and the expression of ISGs, including RyDEN, IFITM-1, OAS-1, ISG15, ISG20, RSAD2, and MxA, were measured using qRT-PCR. The expression of all ISGs after stimulation with IFN-α (100 units), CpG Sup 1;10, and LCP Sup 1:10 compared to Neg Sup 1:10 are shown in Figure 4. Relative to untreated Huh-7 cells, the treatment of IFN-α (100 units), CpG Sup 1:10, and LCP Sup 1:10 increased the ISG expression of IFITM-1, ISG15, ISG20, Mx-A, OAS-1, RSAD2, and RyDEN gene (Figure 4, Appendix A). The nonparametric Steel test revealed a significant increase (*p* < 0.05) in the expression of all tested ISGs in Huh-7 cells after 24 h of LCP Sup 1:10 treatment, except for OAS-1 and RyDEN.

### 3.5. Blockage of Type I IFN Signaling Pathway Minimized the Anti-DENV Effects of LCP Sup

In this study, the effects of blocking human IFN-alpha/beta receptor chain 2 (IFNAR2) on the antiviral effects of LCP Sup on DENV replication were determined. The blocking of IFNAR2 in Huh-7 cells before LCP Sup stimulation significantly increased (*p* < 0.05) the DENV RNA copy number from 105.4 ± 148.2 to 28,992 ± 1057 (Figure 5), representing a 27.44-fold increase. IgG2a antibody was used as an isotype control in the experiment, which did not interfere with either the antiviral effect of LCP Sup or the blocking of IFNAR2.

## 4. Discussion

The previous screening of 125 LAB strains revealed that only *Lactococcus lactis* strain plasma (LC-Plasma), also known as *Lactococcus lactis* subsp. lactis JCM5805, significantly induces IFN-α production from bone marrow (BM)-derived Flt-3L-induced DC cultures [12]. Microscopic observations found that LC-Plasma can be internalized and incorporated into BM-derived Flt-3L-induced DC, resulting in the increased production of IFN-α. This finding is intriguing, because the pDC subset rarely uptakes whole bacteria, possibly due to their large shape and size [34]. Since the IFN-α signaling pathway is the first line of antiviral innate immunity, we sought to determine whether LC-Plasma could be effective against broad-spectrum viral infections. To evaluate the immunomodulatory and antiviral effects of LC-Plasma on humans, PBMC culture supernatant was used as a source of humoral factors for antiviral assays.

Our current results demonstrated that heat-killed LC-Plasma-stimulated PBMCs produced higher IFN-α levels than nonstimulated PBMCs. A similar outcome was observed when LC-Plasma alone, as opposed to other LABs, significantly increased IFN-α production in murine bone marrow dendritic cells (BM-DC). The increase in IFN-α was comparable to the effect seen with the representative TLR9 ligand CpG ODN 1585 [14]. Additionally, a previous study reported that the CpG motif copy number in LC-Plasma positively correlated with pDC stimulatory activity, and the effect of LC-Plasma on pDC activation was dependent on the TLR9-MyD88 signaling pathway [35]. Another study demonstrated no significant difference in IFN-α stimulation between live and heat-killed LC-Plasma, suggesting that the stimulatory effect is derived from a bacterial component, likely the CpG motifs in the LC-Plasma genome, which remain intact and functional even after heat inactivation [12]. Consequently, CpG ODN 2216 (1 µM), which is a human TLR9 agonist, was used as a positive control in this study. It was shown that CpG-stimulated PBMCs produced higher IFN-α levels than nonstimulated PBMCs. This suggested that LC-Plasma and CpG ODN 2216 might similarly stimulate PBMCs to produce the antiviral effects.

Considering LC-Plasma was suggested to activate pDCs via the TLR9/MyD88 pathway [35], and given that pDCs are the primary producer of Type 1 IFN-α [20], this study evaluated the cell surface activation markers HLA-DR and CD86 of pDCs in the PBMC population. The findings showed that LC-Plasma enhanced the expression level of HLA-DR and CD86 in pDCs, suggesting that LC-Plasma could activate pDCs. Similar results were observed in clinical trials, reporting that the oral administration of LC-Plasma for 14 days could increase CD86 and HLA-DR expression, thus reducing the number of incidences (days) of upper respiratory tract infection in male athletes undergoing high-intensity exercise [31]. HLA-DR, also known as major histocompatibility complex (MHC) class 2, is a cell surface receptor on antigen-presenting cells (APCs), which modulates T-helper cell responses [36]. Upon activation, immature DCs undergo maturation and improve their ability to form and accumulate peptides, MHC class II molecules, along with costimulatory molecules, like CD40, CD80, and CD86 [37]. Consequently, HLA-DR is recognized as a marker of DC maturation. This implies that LC-Plasma can potentially facilitate pDC maturation and could be particularly beneficial for the elderly, as they are known to have lower pDC counts and are more susceptible to viral infections [38]. Since HLA-DR on pDCs were upregulated by LC-Plasma, there is a possibility that crosstalk may occur between pDCs and other immune cells, such as myeloid dendritic cells and T cells, directly or in a paracrine fashion through Type 1 IFNs [39]. As a result, the immunomodulatory effects of LC-Plasma might not be limited to pDCs, implying a broader activation that could intricately modulate various immune cell subsets, thereby influencing the overall immune response. Overall, our results suggested that LC-Plasma activates and induces phenotypic maturation of pDCs and, thus, enhances the IFN-α production, which might have a protective effect against viral infection.

Based on LC-Plasma’s ability to stimulate pDCs and promote the production of IFN-α, we postulate that it may have a broad-spectrum antiviral effect against various viruses. The present study evaluates the antiviral effects of LC-Plasma against three important arboviruses, DENV, CHIKV, and ZIKV, which have exhibited an expansion in their geographical distribution, leading to regional transmission and notable outbreaks across all countries in the tropical and subtropical regions [1]. The in vitro infection model for DENV, CHIKV, and ZIKV was established using human liver cells (Huh-7 cells), with PBMC culture supernatant as a source of humoral factors. The results demonstrated that the humoral factors induced by LC-Plasma exhibited a broad-spectrum antiviral effect against DENV, CHIKV, and ZIKV in a dose-dependent manner. Similarly, it was previously reported that the supernatant derived from murine pDCs stimulated with LC-Plasma inhibited DENV replication in a dose-dependent manner [14]. The antiviral properties observed in LCP Sup could be attributed to the elevated levels of IFN-α present in the PBMC culture supernatant. Given that IFN-α is a nonspecific and species-independent component of innate immunity, it may account for its broad antiviral activity against DENV, CHIKV, and ZIKV. A previous study also reported that oral administration of LC-Plasma can stimulate pDCs, which results in the increased production of antiviral factors and reduces the viral load in DENV-infected mice [40].

In a previous study, blocking IFNAR2 abrogated the antiviral effects of the supernatant from LC-Plasma-stimulated murine pDCs against DENV [14]. Conversely, blocking IFN lambda receptor (IFNLR) did not produce a similar outcome, suggesting that the antiviral effects of LC-Plasma depend on the Type I IFN signaling pathway rather than the Type III IFN signaling pathway [14]. In the current study, the suppression of human IFNAR2 significantly negates the antiviral effects of IFN-α produced by LC-Plasma-stimulated PBMCs on DENV replication, leading to a significant increase in DENV RNA copy numbers. These findings further support the hypothesis that the prophylactic actions of LC-Plasma are mediated by the Type I IFN signaling pathway.

Since IFN-α was suggested to be the primary humoral factor responsible for LC-Plasma’s antiviral effects, this study assessed ISGs in Huh-7 cells after a 24-hour stimulation with LCP Sup. ISGs are essential components in the body’s innate immune response, exhibiting diverse activities against viral infections by upregulating their expression in stimulated cells [22]. These proteins target different stages of the virus life cycle, such as obstructing viral entry (IFITM-1) [41], promoting viral RNA degradation (OAS-1) [42] via ribonuclease L, and disrupting lipid metabolism essential for viral replication (RSAD-2) [43]. Collectively, ISGs enhance innate antiviral defenses by inhibiting the replication and facilitating the clearance of infected cells. This study demonstrated that LC-Plasma-induced humoral factors upregulated the expression of these ISGs in Huh-7 cells, thus preventing the replication of DENV, CHIKV, and ZIKV. We found that, while the intrinsic expression of RSAD2 and Mx-A in Huh-7 cells was relatively low, their expression increased substantially upon treatment with IFN-α, CpG Sup 1:10, and LCP Sup 1:10, surpassing the expression of other ISGs tested. This indicates that the treatments might have a common mode of action, specifically targeting the replication stage of viral infection.

Based on our current findings, while LC-Plasma stimulation notably increased IFN-α production in PBMCs compared to CpG ODN stimulation, the expression of ISGs in Huh-7 cells and the antiviral effects of LCP Sup treatment were considerably lower compared to CpG Sup treatment. This difference could be attributed to the potentially distinct IFN-α subtypes produced by PBMCs upon LC-Plasma or CpG ODN treatment. Although all IFN-α subtypes bind the same receptor (IFNAR), they exhibit different binding affinities (*K*_D_ values) and biological activities [44]. For instance, IFN-α2a and IFN-α1 had lower affinities for the IFNAR compared to other subtypes, which is associated with the anti-proliferative potency of the different IFN-α subtypes [44,45]. Variations in receptor affinities might lead to different downstream signaling cascades, influencing the phosphorylation of signal transducer and activator of transcription (STAT) molecules and mitogen-activated protein kinase (MAPK) [46]. Moreover, the cell type responsible for IFN production might also influence the expression and action of different IFN-α subtypes [47]. Both LC-Plasma and CpG ODN enhance IFN-α production, but they may stimulate different cell types in PBMCs to produce IFN-α. Additionally, as a whole microorganism, LC-Plasma has the potential to stimulate IFNα production through multiple cell types. It was reported that less potent antiviral IFN-α subtypes (IFNα1, IFNα6, IFNα16, and IFNα10) exhibited comparatively lower expression values of specific ISGs; whereas, potent antiviral IFNα subtypes induced higher ISG expression [48]. Thus, LC-Plasma treatment might stimulate less potent antiviral IFN-α subtypes, while CpG ODN might stimulate more potent antiviral IFNα subtypes, which might explain the lower antiviral effect and ISG expression upon LCP Sup treatment compared to CpG Sup.

Despite LC-Plasma Sup showing less potent antiviral effects compared to CpG Sup or recombinant IFN-α, its use remains promising for viral infection management. Exogenous IFN-α therapy or CpG ODN intervention, especially when administered late, has been linked to prolonged hospitalization, and improper or delayed IFN-α treatment in viral infections can lead to hyper-inflammation and worsen outcomes, especially in older adults or individuals with high viral exposure [17]. Genetic factors, such as mutations in IFN pathways or the presence of neutralizing autoantibodies, can further complicate IFN-α effectiveness and contribute to severe disease. Moreover, endogenous IFN levels, supported by a balanced commensal microbiota, are crucial for innate immunity. Dysbiosis, often caused by antibiotic use, can reduce IFN signals and compromise antiviral defense. While exogenous IFN-α treatment can be effective against viruses, like Hepatitis C, it carries risks of adverse effects, including flu-like symptoms, gastrointestinal issues, and hematologic toxicity. These concerns make the prophylactic use of IFN-α unviable due to safety and cost considerations. Alternatively, LC-Plasma offers a safer immune modulation approach, effectively activating pDCs without adverse events. It is safe even at high doses in adults and young children, presenting a viable strategy for preventing virus infection [17].

Combination therapy using probiotic LC-Plasma alongside other antiviral agents could be a promising strategy to enhance antiviral efficacy [49]. LC-Plasma, known for activating plasmacytoid dendritic cells (pDCs) and boosting type I interferon production, could provide a strong immunomodulatory response. By combining LC-Plasma with other antivirals, such as monoclonal antibodies, small-molecule antivirals, or repurposed drugs, a multifaceted approach could be developed to inhibit various stages of the viral life cycle. For instance, using LC-Plasma with monoclonal antibodies that neutralize the virus could create a synergistic effect, where LC-Plasma primes the immune system, while antibodies block viral entry. Similarly, when combined with small-molecule inhibitors that prevent viral replication, LC-Plasma’s ability to enhance innate immunity could lead to a more comprehensive antiviral defense. This approach could also potentially mitigate the risk of viral resistance, as the virus would face multiple inhibitory mechanisms simultaneously [49]. In addition, the combination therapy could be particularly beneficial in populations with weakened immune responses, such as the elderly or those with underlying conditions, where boosting innate immunity is crucial. Given the safety profile of LC-Plasma and its potential for widespread use, further research is warranted to explore these synergies and optimize dosing regimens for maximum clinical benefit. Furthermore, future research directions could be expanded to clinical studies or in vivo models to test the efficacy of LC-Plasma in preventing arboviral infections. Investigating the potential of LC-Plasma in these settings could pave the way for new preventive strategies, especially in endemic regions.

## 5. Conclusions

This study has demonstrated that LC-Plasma induces a broad-spectrum antiviral effect against DENV, CHIKV, and ZIKV, primarily by enhancing type I IFN production. This increased IFNα expression is attributed to the activation of pDCs, as evidenced by the upregulation of activation markers HLA-DR and CD86 within the pDC population in PBMCs. These increased IFN-α levels are likely responsible for the enhanced expression of ISGs in Huh-7 cells, resulting in a broad-spectrum antiviral effect of LC-Plasma. These findings highlighted the potential of LC-Plasma as a possible prophylactic option against viral infections. While previous antiviral studies have focused on small chemical molecules, peptides, or natural compounds [29,49,50,51], there are many limitations on their applications, including those related to effectivity, bioavailability, accessibility, safety, and cost [52]. In contrast, heat-killed LC-Plasma that targets the host immune response does not encounter these issues, warranting further in vivo investigation either using a mouse model or conduct rigorous clinical trials in both healthy individuals and those with viral infections.

## Figures and Tables

**Figure 1 microorganisms-12-02304-f001:**
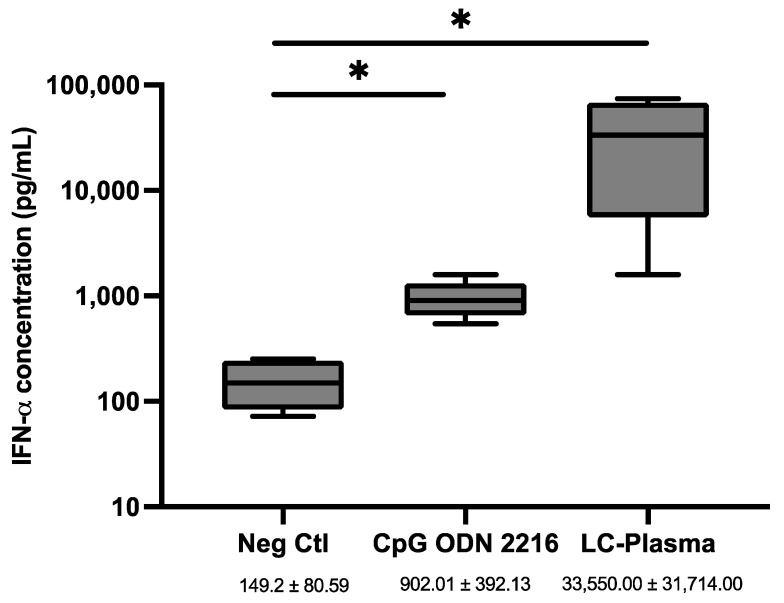
The concentration of IFN-α in the supernatant of PBMCs stimulated with CpG ODN2216 (1 µM) and LC-Plasma (10 µg/mL). Box and violin graph illustrating the median ± SD (N = 5) of IFN-α concentration in the PBMC supernatant determined using ELISA. Neg Ctl = negative control, indicating PBMCs without treatment. * *p* < 0.05 (nonparametric Kruskal–Wallis and Steel test).

**Figure 2 microorganisms-12-02304-f002:**
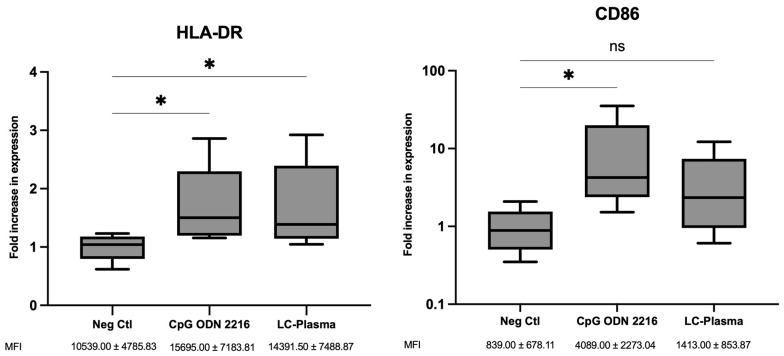
Fold increase in expression of CD86 and HLA-DR of pDCs after 24 h of CpG ODN 2216 (1 µM) and LC-Plasma (10 µg/mL) stimulation. Box and violin graph illustrating the median ± SD (N = 5) fold increase in expression of HLR-DR (**left**) and CD86 (**right**) of pDCs without stimulation (Neg Ctl) and after CpG ODN 2216 and LC-Plasma stimulation. * *p* < 0.05, ns= non-significant difference (nonparametric Kruskal–Wallis and Steel test).

**Figure 3 microorganisms-12-02304-f003:**
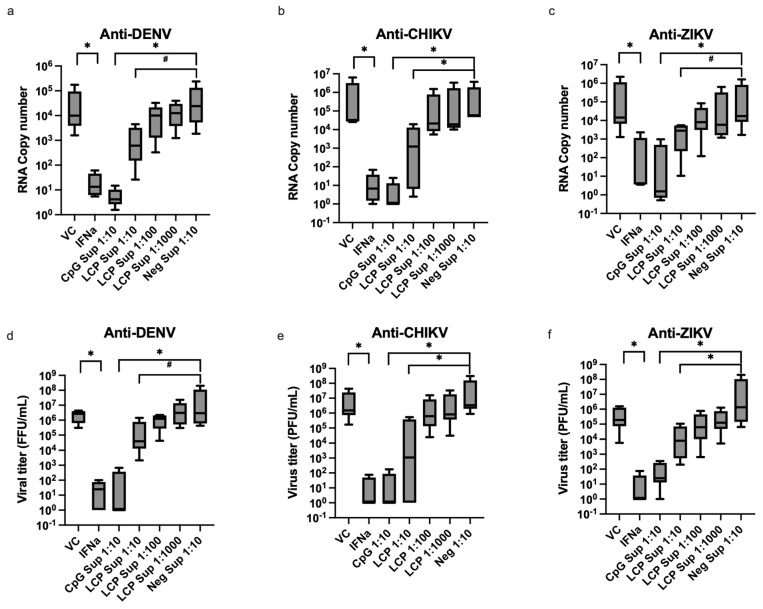
Antiviral effects of supernatant from PBMCs stimulated with LC-Plasma against DENV, CHIKV, and ZIKV replication. Comparison of DENV, CHIKV, and ZIKV titers from 10-fold diluted supernatant of nonstimulated PBMC (Neg Sup 1:10), 10-fold diluted supernatants of LC-Plasma-stimulated PBMC (LCP Sup 1:10), and 10-fold diluted supernatants of CpG ODN 2216-stimulated PBMC (CpG Sup 1:10). The antiviral effects of LCP Sup 1:10 against DENV, CHIKV, and ZIKV replication were observed via qRT-PCR (**a**–**c**) and the infectious virus reduction assay (plaque/foci assay) (**d**–**f**) The Box and violin graphs represent the median ± SD (N = 5). * *p* < 0.05 indicates a significant difference, while # *p* < 0.1 indicates no significant difference but showed a tendency toward reduction (nonparametric Kruskal–Wallis and Steel test).

**Figure 4 microorganisms-12-02304-f004:**
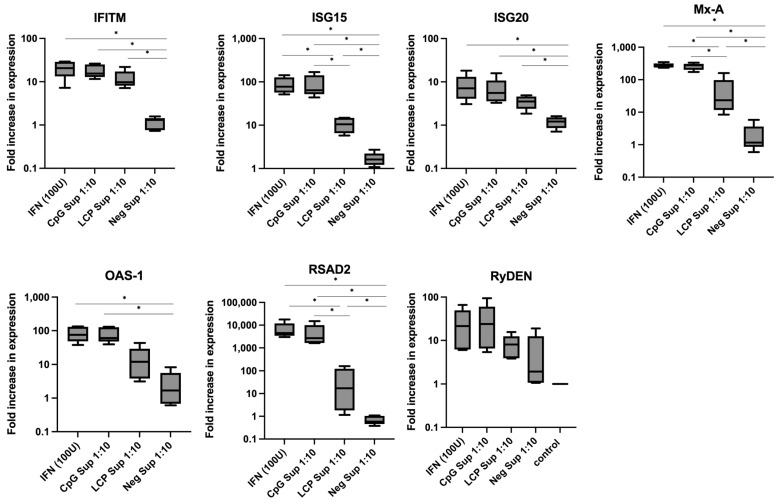
Induction of IFN-stimulated gene (ISG) expression in Huh-7 cells. Huh-7 cells were treated with recombinant IFN-α (100u), CpG Sup 1:10, LCP Sup 1:10, or Neg Sup 1:10, and the expression of ISGs was measured using qRT-PCR. The comparison included IFITM, ISG15, ISG20, MxA, OAS-1, RSAD2, and RyDEN expression in Huh-7 cells following treatments with recombinant IFN-α, CpG Sup 1:10, and LCP Sup 1:10. The gene expression levels were referenced to the untreated Huh-7 cells (cell control) and normalized using the housekeeping gene beta-actin. The Box and violin graph represents median ± SD (N = 5). * *p* < 0.05 indicates statistical significance (nonparametric Kruskal–Wallis and Steel tests).

**Figure 5 microorganisms-12-02304-f005:**
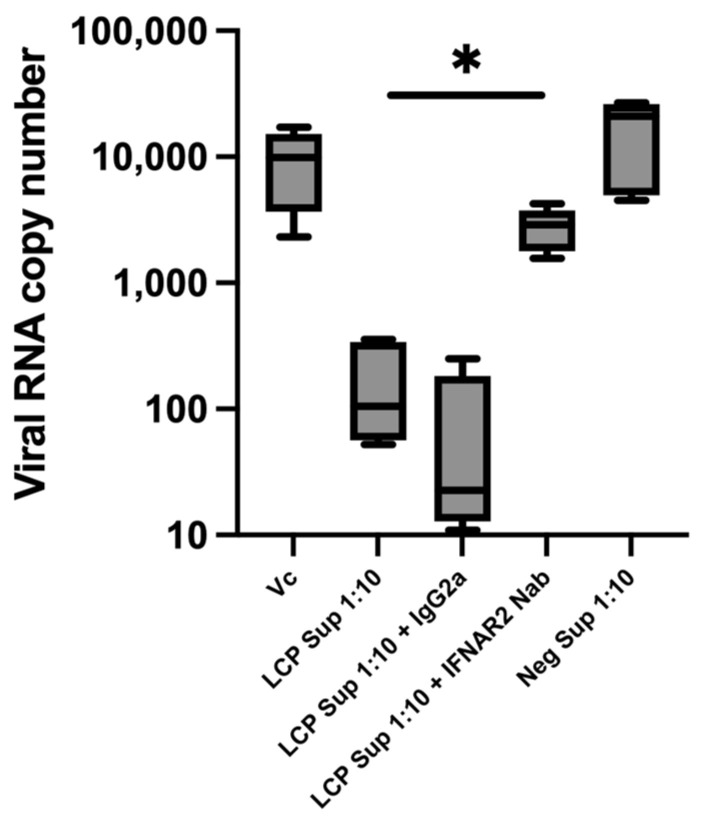
Effects of blocking Type I IFN signaling. To further elucidate the mechanism of the anti-DENV effect of LC-Plasma, Huh-7 cells were treated with blocking antibodies against IFNAR2 for 1 h before stimulation with 10-fold diluted LCP Sup (LCP Sup 1:10). The Box and violin graph represents median ± SD (N = 5). * *p* < 0.05 indicates statistical significance (nonparametric Kruskal–Wallis and Steel tests).

## Data Availability

The original contributions presented in the study are included in the article; further inquiries can be directed to the corresponding authors.

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
