# Peer review of "The Antiviral Effects of Heat-Killed Lactococcus lactis Strain Plasma Against Dengue, Chikungunya, and Zika Viruses in Humans by Upregulating the IFN-α Signaling Pathway"

_microorganisms, 2024, doi:10.3390/microorganisms12112304_

Round 1
Reviewer 1 Report
Comments and Suggestions for Authors
In the manuscript “Heat-killed Lactococcus Lactis Strain Plasma Stimulates PBMCs to Elicit Interferon-Alpha Production and Promote Broad-Spectrum Antiviral Activity Against Dengue, Chikungunya, and Zika Viruses”, the authors conducted this study to assess the immunomodulatory effects of heat-killed Lactococcus lactis strain Plasma (LC-Plasma) on peripheral blood mononuclear cells (PBMCs) and its subsequent antiviral response against Dengue virus (DENV), Chikungunya virus (CHIKV) and Zika virus (ZIKV). They highlighted the potential of LC-Plasma as a possible prophylactic or therapeutic option against viral infections by upregulating the IFN-α signaling pathway.
The manuscript is generally well-addressed and well-cited; however, I have some comments/suggestions:
Line 5: Please rewrite the title to be more representative and specific. I suggest being: The antiviral effects of heat-killed Lactococcus Lactis Strain Plasma against Dengue, Chikungunya, and Zika Viruses in human by upregulating the IFN-α signaling pathway.
Line 6: Please note that author "Vunjia Tiong" is not corresponding author to add symbol * to his/her name. You mentioned the last author below at line 14 as the only corresponding author.
Line 17: Please mention by names what do you mean by "these virus infections". Please be specific.
Line 22: Before start demonstrating your findings at the sentence of " LC-Plasma-stimulated PBMCs demonstrated elevated", please describe briefly the main methods or treatments applied. Include any relevant preregistration numbers, and species and strains of probiotics used.
Line 23: Please verify the "CD86 and HLA-DR" as they are first time mentioned at the manuscript.
Line 51: The paragraph starting from line 41 by "This expansion has " until here on line 51, is too long. Please rewrite and focus.
Line 73: I think you don't need to mention other more types of probiotics starting from line 70 until line 73, as you did not mention the targeted one in your manuscript "Lactococcus lactis strain Plasma (LC-Plasma)", just to be more close and species specific.
Line 90: I suggest moving the above paragraph starting from line 75 by " In combating viral infections " to be AFTER the next paragraph starting by "Lactococcus lactis strain Plasma (LC-Plasma " on line 92. Just give a good sequencing of the ideas and easy tracking the literature.
Line 117: please add the year of this study approval.
Line 130: Please add the company and country producing this trypan blue stain.
Line 135: Please add the source of Vero cells.
Line 146: Please add the source of previous mentioned viruses.
Line 239: Please add the catalog number for this kit and other kits used in your study along the manuscript.
Line 239: Please add the catalog number for this kit and other kits used in your study along the manuscript.
Line 490: I suggest initiating a separate section with a bold title of conclusion instead of this paragraph started by " In conclusion,"
Line 512: Through the previous "In conclusion paragraph" that started from line 490, please rewrite the conclusion to be more specific and focus. It is too long. It should summarize the key findings of your study, reiterate the main points of your argument, highlight the significance of your research, and discuss potential implications and future research directions while avoiding introducing new information.
Line: 529: please add the year of approval beside this grant number.
References # 1, 2, 10, 13, 17, 21 and 44 are incomplete, specially the page numbers. Please revise and follow the journal guidelines.
Comments on the Quality of English LanguageMinor editing of English language is required.
Author Response
Comments |
Response and amendment |
Page and line number |
Line 5: Please rewrite the title to be more representative and specific. I suggest being: The antiviral effects of heat-killed Lactococcus Lactis Strain Plasma against Dengue, Chikungunya, and Zika Viruses in human by upregulating the IFN-α signaling pathway. |
Thank you for the suggestion. We have revised the title to "The Antiviral Effects of Heat-Killed Lactococcus lactis Strain Plasma Against Dengue, Chikungunya, and Zika Viruses in Humans by Upregulating the IFN-α Signaling Pathway," making it more representative and specific. |
Page 1 line 1-3 |
Line 6: Please note that author "Vunjia Tiong" is not corresponding author to add symbol * to his/her name. You mentioned the last author below at line 14 as the only corresponding author. |
We appreciate the correction. The symbol (*) for the corresponding author has been added to "Vunjia Tiong" |
Page 1 line 5 |
Line 17: Please mention by names what do you mean by "these virus infections". Please be specific. |
We have clarified "these virus infections" by specifying DENV, CHIKV, and ZIKV. Thank you for pointing this out. |
Page 1 line 16 |
Line 22: Before start demonstrating your findings at the sentence of " LC-Plasma-stimulated PBMCs demonstrated elevated", please describe briefly the main methods or treatments applied. Include any relevant preregistration numbers, and species and strains of probiotics used. |
We have included a brief description of the main methods and treatments, along with relevant preregistration details and information about the probiotic species and strains used.
In the manuscript: “The present study aimed to assess the immunomodulatory effects of heat-killed Lactococcus lactis strain Plasma (LC-Plasma), JCM5805 on peripheral blood mononuclear cells (PBMCs) and its subsequent antiviral response against DENV, CHIKV, and ZIKV. To evaluate the immunomodulatory effects of LC-Plasma on PBMCs isolated from healthy individual, PBMCs were cultured at a density of 2×105 cells/well and stimulated with 10 µg/mL of LC-Plasma.” |
Page 1 line 21-23 |
Line 23: Please verify the "CD86 and HLA-DR" as they are first time mentioned at the manuscript. |
We have verified the mention of "CD86 and HLA-DR" and added a brief explanation to introduce these markers, ensuring clarity for the readers.
In the manuscript: “LC-Plasma-stimulated PBMCs demonstrated elevated interferon-alpha (IFN-α) production and Cluster of Differentiation 86 (CD86) and Human Leukocyte Antigen-DR isotype (HLA-DR) upregulation, potentially linked to plasmacytoid dendritic cells (pDCs) activation.” |
Page 1 line 24-25 |
Line 51: The paragraph starting from line 41 by "This expansion has " until here on line 51, is too long. Please rewrite and focus. |
We have shortened and refocused the paragraph starting from line 41 to line 51 for better readability and conciseness. |
Page 1-2 Line 43-66 |
Line 73: I think you don't need to mention other more types of probiotics starting from line 70 until line 73, as you did not mention the targeted one in your manuscript "Lactococcus lactis strain Plasma (LC-Plasma)", just to be more close and species specific. |
We have removed the additional types of probiotics mentioned from line 70 to line 73 to maintain specificity and align with our focus on Lactococcus lactis strain Plasma (LC-Plasma). |
NA |
Line 90: I suggest moving the above paragraph starting from line 75 by " In combating viral infections " to be AFTER the next paragraph starting by "Lactococcus lactis strain Plasma (LC-Plasma " on line 92. Just give a good sequencing of the ideas and easy tracking the literature. |
We have reordered the paragraphs for improved sequencing, placing the paragraph on viral infection mechanisms after the discussion of LC-Plasma on line 92, as suggested. |
Page 2-3 line 103 |
Line 117: please add the year of this study approval. |
We have added the year of the study approval to provide complete information. In the manuscript: “The Medical Research Ethics Committee of Universiti Malaya approved this study in 2022 (UMMC MREC No. 202256-11216).” |
Page 3 Line 161 |
Line 130: Please add the company and country producing this trypan blue stain. |
We have included the company and country of the trypan blue stain supplier.
In the manuscript: “PBMCs viability was evaluated using trypan blue (Bio-rad, USA) staining and the per-centage of viability was calculated using a TC20 automated cell counter (Bio-rad, USA). Upon thawing, PBMCs viability was 80% and above.” |
Page 3 Line 175 |
Line 135: Please add the source of Vero cells. |
The source of the Vero and Huh-7 cells has been added for proper attribution.
In the manuscript: “Vero cells (African Green Monkey kidney) (ECACC strain) were cultured and maintained in high glucose (4g/L) Dulbecco's Modified Eagle Medium (DMEM) (Corning, USA). In contrast, Huh-7 cells (JCRB Cell Bank, catalog number: JCRB0403) were cultured and maintained in low glucose (1g/L) DMEM (Corning, USA).” |
Page 3 line 180-182 |
Line 146: Please add the source of previous mentioned viruses. |
We have included the source details for the viruses used in the study.
In the manuscript: “DENV-2 (New Guinea C (NGC) strain, accession number: ATCC VR-1584), CHIKV (ECSA genotype, MY/065/08/ accession number: FN295485), and ZIKV (strain P6740, obtained from the University of Texas Medical Branch).” |
Page 3 Line 191-192 |
Line 239: Please add the catalog number for this kit and other kits used in your study along the manuscript. |
We have added the catalog numbers for the kits used in the study to ensure full transparency and reproducibility.
In the manuscript: 1. Human IFN-α ELISA Kit (Elabscience, China, E-EL-H6125) 2. QIAampR Viral RNA extraction kit (QIAGEN, Germany, QIAG-52904) 3. SensiFast SYBER Hi-ROX one-step kit (Bioline, UK, BIO-73005) 4. RNeasy Kit (Qiagen, Germany, QIAG-74106) 5. iScript cDNA synthesis kit (BioRad, USA, 1708891) 6. SYBR green-based Luna® Universal qRT-PCR Master Mix (NEB, USA, M3003L)” |
Line 243, 292, 295, 305, 306, 308, |
Line 490: I suggest initiating a separate section with a bold title of conclusion instead of this paragraph started by " In conclusion," |
We have created a new section titled "Conclusion" and moved the relevant paragraph to this section, as suggested.
|
Page 13 line 599-611 |
Line 512: Through the previous "In conclusion paragraph" that started from line 490, please rewrite the conclusion to be more specific and focus. It is too long. It should summarize the key findings of your study, reiterate the main points of your argument, highlight the significance of your research, and discuss potential implications and future research directions while avoiding introducing new information. |
The conclusion has been rewritten to be more concise and focused. It now summarizes key findings, emphasizes the significance of our research, and outlines future directions without introducing new information. |
Page 13 line 599-611 |
Line: 529: please add the year of approval beside this grant number. |
We have added the year of approval beside the grant number to provide complete funding information. |
Page 14 line 692 |
References # 1, 2, 10, 13, 17, 21 and 44 are incomplete, specially the page numbers. Please revise and follow the journal guidelines. |
We have revised references #1, 2, 10, 13, 17, 21, and 44, ensuring all entries, including page numbers, adhere to the journal’s guidelines. |
References |
Reviewer 2 Report
Comments and Suggestions for Authors
The manuscript "Heat-killed Lactococcus Lactis Strain Plasma Stimulates PBMCs to Elicit Interferon-Alpha Production and Promote Broad-Spectrum Antiviral Activity Against Dengue, Chikungunya, and Zika Viruses" investigates the antiviral efficacy of heat-killed Lactococcus lactis strain Plasma (LC-Plasma) in stimulating interferon-alpha (IFN-α) production in peripheral blood mononuclear cells (PBMCs). The study examines the effects of this probiotic-derived component against tropical viruses—Dengue virus (DENV), Chikungunya virus (CHIKV), and Zika virus (ZIKV). Findings reveal that LC-Plasma-stimulated PBMCs release IFN-α and induce interferon-stimulated genes (ISGs), resulting in broad-spectrum antiviral responses.
The manuscript "Heat-killed Lactococcus Lactis Strain Plasma Stimulates PBMCs to Elicit Interferon-Alpha Production and Promote Broad-Spectrum Antiviral Activity Against Dengue, Chikungunya, and Zika Viruses"investigates the antiviral efficacy of heat-killed Lactococcus lactis strain Plasma (LC-Plasma) in stimulating interferon-alpha (IFN-α) production in peripheral blood mononuclear cells (PBMCs). The study examines the effects of this probiotic-derived component against tropical viruses—Dengue virus (DENV), Chikungunya virus (CHIKV), and Zika virus (ZIKV). Findings reveal that LC-Plasma-stimulated PBMCs release IFN-α and induce interferon-stimulated genes (ISGs), resulting in broad-spectrum antiviral responses.
Major issues:
Critical Discussion of Results:
The manuscript would benefit from a deeper critical evaluation of the findings. For example, while LC-Plasma stimulates IFN-α production, the potential limitations of using heat-killed organisms compared to live probiotics in antiviral immunity should be discussed. Heat-killed probiotics may lack some immune-activating components present in live strains, which could limit their efficacy. Additionally, the lower antiviral effect of LCP-Sup compared to CpG-Sup should be addressed. This difference could stem from varying IFN-α subtypes produced upon LC-Plasma and CpG treatment, impacting potency and effectiveness. A discussion on these nuances could offer insight into why LC-Plasma alone may or may not serve as a standalone antiviral treatment.
Discussion:
The discussion is informative but would benefit from a more critical perspective. For instance, while the results indicate LC-Plasma’s potential for broad-spectrum antiviral effects, addressing why its effects may be less potent than other treatments would provide a balanced view. The difference in efficacy between LCP-Sup and CpG-Sup also warrants deeper discussion, particularly considering the differences in IFN-α subtype activation. Additionally, future research directions could be expanded. Suggestions for clinical studies or in vivo models to test LC-Plasma in preventing arboviral infections would be valuable. Discussing the potential for combination therapies, where LC-Plasma is used alongside antiviral drugs, could highlight its role in integrative treatment approaches.
Minor issues:
Linne 26: The expression of IFN-stimulated genes (ISGs) expression..." Correction: Remove redundant "expression."
Line 32: “Thus, LC-Plasma may serve as a rational adjunctive treatment for ameliorate viral diseases, warranting future clinical trials.” should be corrected to “treatment to ameliorate
Line 39: "...due to the abundance presence of mosquitoes in these areas [1]." should be "...due to the abundance of mosquitoes in these areas [1]."
Line 43-44: please rephrase for better flow the following sentence: "Currently, 2.5 billion people worldwide are at risk for DENV infection, with a global annual infection of an estimated 390 million cases, including 500,000 hospitalizations and approximately 25,000 deaths."
Line 65-66: "Probiotics are living microorganisms which both confer health benefits to the host when ingested in adequate amounts [11]." should be "Probiotics are living microorganisms that confer health benefits to the host when ingested in adequate amounts [11]."
Corrected Sentence: "PBMCs were isolated from whole blood using a BD Vacutainer CPT glass molecular diagnostics tube, according to the manufacturer’s protocol."
Line 123: "PBMCs were washed twice with phosphate-buffered saline (PBS) and aliquots of the PBMCs pellet were resuspended in Cellbanker2 (ZENOGEN, Japan) and cryopreserved at −80∘C Upon thawing, PBMCs were cultured..." should be "PBMCs were washed twice with phosphate-buffered saline (PBS), and aliquots of the PBMCs pellet were resuspended in Cellbanker2 (ZENOGEN, Japan) and cryopreserved at −80∘C. Upon thawing, PBMCs were cultured..."
Line 152-153: "The Vero cells were seeded overnight at a density of 1×105cells/well in 24 well plates." should be "Vero cells were seeded overnight at a density of 1×105cells/well in 24-well plates."
Line 153: "... monolayer confluent Vero cells were infected with ten-fold serial diluted DENV, CHIKV, and ZIKV for one hour at 37∘C fpr preadsorption." should be "... monolayer confluent Vero cells were infected with ten-fold serial diluted DENV, CHIKV, and ZIKV for one hour at 37∘C for preadsorption."
Line 171: "Plaques in each well was counted, and viral titer was determined in plaque-forming unit per microliter (PFU/mL) [36]." should be "Plaques in each well were counted, and viral titer was determined in plaque-forming unit per microliter (PFU/mL) [36]."
Line 287: "There was a statistically significant (non-parametric Kruskal-Wallis and Steel test; p<0.05) increase in the expression level of HLA-DR due to LC-Plasma treatment." should be "There was a statistically significant (non-parametric Kruskal-Wallis and Steel test; p<0.05) increase in the expression level of HLA-DR due to LC-Plasma treatment."
Sentence: "...the increased in CD86 levels in pDCs following LC-Plasma treatment..." Correction: "increased" should be "increase."
Line 387: “the increased of IFN-α” should be corrected to "the increase in IFN-α"
"The increased of IFN-a was comparable to the effect seen with the representative TLR9 ligand CpG ODN 1585 [29]." should be "The increase in IFN-a was comparable to the effect seen with the representative TLR9 ligand CpG ODN 1585 [29]."
Line 394: "This suggested that LC-Plasma and CpG ODN 2216 might similarly stimulate PBMCs to produce the antiviral effects.[44]" should be "This suggested that LC-Plasma and CpG ODN 2216 might similarly stimulate PBMCs to produce the antiviral effects [44]."
Line 470: "This difference could be attributed to the potentially distinct IFNa subtypes produced by PBMCs upon LC-Plasma or CpG ODN treatment." should be "This difference could be attributed to the potentially distinct IFN-a subtypes produced by PBMCs upon LC-Plasma or CpG ODN treatment."The manuscript "Heat-killed Lactococcus Lactis Strain Plasma Stimulates PBMCs to Elicit Interferon-Alpha Production and Promote Broad-Spectrum Antiviral Activity Against Dengue, Chikungunya, and Zika Viruses"investigates the antiviral efficacy of heat-killed Lactococcus lactis strain Plasma (LC-Plasma) in stimulating interferon-alpha (IFN-α) production in peripheral blood mononuclear cells (PBMCs). The study examines the effects of this probiotic-derived component against tropical viruses—Dengue virus (DENV), Chikungunya virus (CHIKV), and Zika virus (ZIKV). Findings reveal that LC-Plasma-stimulated PBMCs release IFN-α and induce interferon-stimulated genes (ISGs), resulting in broad-spectrum antiviral responses.
Major issues:
Critical Discussion of Results:
The manuscript would benefit from a deeper critical evaluation of the findings. For example, while LC-Plasma stimulates IFN-α production, the potential limitations of using heat-killed organisms compared to live probiotics in antiviral immunity should be discussed. Heat-killed probiotics may lack some immune-activating components present in live strains, which could limit their efficacy. Additionally, the lower antiviral effect of LCP-Sup compared to CpG-Sup should be addressed. This difference could stem from varying IFN-α subtypes produced upon LC-Plasma and CpG treatment, impacting potency and effectiveness. A discussion on these nuances could offer insight into why LC-Plasma alone may or may not serve as a standalone antiviral treatment.
Discussion:
The discussion is informative but would benefit from a more critical perspective. For instance, while the results indicate LC-Plasma’s potential for broad-spectrum antiviral effects, addressing why its effects may be less potent than other treatments would provide a balanced view. The difference in efficacy between LCP-Sup and CpG-Sup also warrants deeper discussion, particularly considering the differences in IFN-α subtype activation. Additionally, future research directions could be expanded. Suggestions for clinical studies or in vivo models to test LC-Plasma in preventing arboviral infections would be valuable. Discussing the potential for combination therapies, where LC-Plasma is used alongside antiviral drugs, could highlight its role in integrative treatment approaches.
Minor issues:
Linne 26: The expression of IFN-stimulated genes (ISGs) expression..." Correction: Remove redundant "expression."
Line 32: “Thus, LC-Plasma may serve as a rational adjunctive treatment for ameliorate viral diseases, warranting future clinical trials.” should be corrected to “treatment to ameliorate
Line 39: "...due to the abundance presence of mosquitoes in these areas [1]." should be "...due to the abundance of mosquitoes in these areas [1]."
Line 43-44: please rephrase for better flow the following sentence: "Currently, 2.5 billion people worldwide are at risk for DENV infection, with a global annual infection of an estimated 390 million cases, including 500,000 hospitalizations and approximately 25,000 deaths."
Line 65-66: "Probiotics are living microorganisms which both confer health benefits to the host when ingested in adequate amounts [11]." should be "Probiotics are living microorganisms that confer health benefits to the host when ingested in adequate amounts [11]."
Corrected Sentence: "PBMCs were isolated from whole blood using a BD Vacutainer CPT glass molecular diagnostics tube, according to the manufacturer’s protocol."
Line 123: "PBMCs were washed twice with phosphate-buffered saline (PBS) and aliquots of the PBMCs pellet were resuspended in Cellbanker2 (ZENOGEN, Japan) and cryopreserved at −80∘C Upon thawing, PBMCs were cultured..." should be "PBMCs were washed twice with phosphate-buffered saline (PBS), and aliquots of the PBMCs pellet were resuspended in Cellbanker2 (ZENOGEN, Japan) and cryopreserved at −80∘C. Upon thawing, PBMCs were cultured..."
Line 152-153: "The Vero cells were seeded overnight at a density of 1×105cells/well in 24 well plates." should be "Vero cells were seeded overnight at a density of 1×105cells/well in 24-well plates."
Line 153: "... monolayer confluent Vero cells were infected with ten-fold serial diluted DENV, CHIKV, and ZIKV for one hour at 37∘C fpr preadsorption." should be "... monolayer confluent Vero cells were infected with ten-fold serial diluted DENV, CHIKV, and ZIKV for one hour at 37∘C for preadsorption."
Line 171: "Plaques in each well was counted, and viral titer was determined in plaque-forming unit per microliter (PFU/mL) [36]." should be "Plaques in each well were counted, and viral titer was determined in plaque-forming unit per microliter (PFU/mL) [36]."
Line 287: "There was a statistically significant (non-parametric Kruskal-Wallis and Steel test; p<0.05) increase in the expression level of HLA-DR due to LC-Plasma treatment." should be "There was a statistically significant (non-parametric Kruskal-Wallis and Steel test; p<0.05) increase in the expression level of HLA-DR due to LC-Plasma treatment."
Sentence: "...the increased in CD86 levels in pDCs following LC-Plasma treatment..." Correction: "increased" should be "increase."
Line 387: “the increased of IFN-α” should be corrected to "the increase in IFN-α"
"The increased of IFN-a was comparable to the effect seen with the representative TLR9 ligand CpG ODN 1585 [29]." should be "The increase in IFN-a was comparable to the effect seen with the representative TLR9 ligand CpG ODN 1585 [29]."
Line 394: "This suggested that LC-Plasma and CpG ODN 2216 might similarly stimulate PBMCs to produce the antiviral effects.[44]" should be "This suggested that LC-Plasma and CpG ODN 2216 might similarly stimulate PBMCs to produce the antiviral effects [44]."
Line 470: "This difference could be attributed to the potentially distinct IFNa subtypes produced by PBMCs upon LC-Plasma or CpG ODN treatment." should be "This difference could be attributed to the potentially distinct IFN-a subtypes produced by PBMCs upon LC-Plasma or CpG ODN treatment."
Author Response
Comments |
Response and amendment |
Page and line number |
The manuscript would benefit from a deeper critical evaluation of the findings. For example, while LC-Plasma stimulates IFN-α production, the potential limitations of using heat-killed organisms compared to live probiotics in antiviral immunity should be discussed. Heat-killed probiotics may lack some immune-activating components present in live strains, which could limit their efficacy.
For instance, while the results indicate LC-Plasma’s potential for broad-spectrum antiviral effects, addressing why its effects may be less potent than other treatments would provide a balanced view. The difference in efficacy between LCP-Sup and CpG-Sup also warrants deeper discussion, particularly considering the differences in IFN-α subtype activation. |
Thank you for highlighting this important aspect. We agree that a deeper evaluation of the differences between live and heat-killed probiotics is valuable. LC-Plasma retains its capacity to stimulate IFN-α production even after heat inactivation, likely due to the preservation of CpG motifs [7]. While we recognize that heat-killed organisms may lack certain immune-activating components present in live strains, such as metabolic byproducts or cell-surface molecules that could contribute to immune modulation, this appears not to be the case with LC-Plasma. Heat-inactivated LC-Plasma maintains its immunomodulatory activity, offering advantages such as an enhanced safety profile, greater stability, and broader applicability for use in various populations, including immunocompromised individuals.
In the manuscript: “Another study demonstrated no significant difference in IFN-α stimulation between live and heat-killed LC-Plasma, suggesting that the stimulatory effect is derived from a bacterial component, likely the CpG motifs in the LC-Plasma genome, which remain intact and functional even after heat inactivation [7]” |
Page 10 Line 449-453 |
Additionally, the lower antiviral effect of LCP-Sup compared to CpG-Sup should be addressed. This difference could stem from varying IFN-α subtypes produced upon LC-Plasma and CpG treatment, impacting potency and effectiveness. A discussion on these nuances could offer insight into why LC-Plasma alone may or may not serve as a standalone antiviral treatment.
For instance, while the results indicate LC-Plasma’s potential for broad-spectrum antiviral effects, addressing why its effects may be less potent than other treatments would provide a balanced view. The difference in efficacy between LCP-Sup and CpG-Sup also warrants deeper discussion, particularly considering the differences in IFN-α subtype activation. |
Thank you for your suggestion. We agree that discussing the differences in antiviral efficacy between LCP-Sup and CpG-Sup is crucial. And it was already discussed in. While LC-Plasma significantly increased IFN-α production in PBMCs, the antiviral effects and ISG expression were lower compared to CpG-Sup. This may be due to differences in IFN-α subtypes produced. Although all IFN-α subtypes use the same receptor (IFNAR), they vary in binding affinities and biological activities. Weaker subtypes, like IFN-α1, may be produced more by LC-Plasma, leading to less potent antiviral effects. Additionally, LC-Plasma might activate multiple cell types and pathways, contributing to a broader but less targeted response. In contrast, CpG ODN likely stimulates more potent IFN-α subtypes, resulting in stronger antiviral activity. We will include this discussion to explain why LC-Plasma’s effects may differ and highlight its potential role in a broader antiviral strategy.
|
Page 12- Line 535-574 |
Additionally, future research directions could be expanded. Suggestions for clinical studies or in vivo models to test LC-Plasma in preventing arboviral infections would be valuable. |
Future direction has been added. |
Page 13, line 592-594 |
Discussing the potential for combination therapies, where LC-Plasma is used alongside antiviral drugs, could highlight its role in integrative treatment approaches. |
Thank you for your valuable feedback. Based on your suggestion, we have edited the manuscript to emphasize the potential of combination therapy involving LC-Plasma alongside antiviral agents. We have detailed how LC-Plasma, with its known ability to activate plasmacytoid dendritic cells and boost type I interferon production, could complement other treatments. The combination approach could offer a multifaceted antiviral strategy, targeting different stages of the viral life cycle. This approach aims to amplify the overall antiviral efficacy while reducing the risk of viral resistance. Additionally, the safety profile of LC-Plasma makes it a suitable candidate for broader use, particularly in vulnerable populations. We hope these modifications address your concerns and enrich the discussion of LC-Plasma's role in integrative antiviral strategies. |
Page 13, Line 576-596 |
Linne 26: The expression of IFN-stimulated genes (ISGs) expression..." Correction: Remove redundant "expression." |
Minor concerns are corrected as suggested |
Line 28 |
Line 32: “Thus, LC-Plasma may serve as a rational adjunctive treatment for ameliorate viral diseases, warranting future clinical trials.” should be corrected to “treatment to ameliorate |
Minor concerns are corrected as suggested |
Line 34 |
Line 39: "...due to the abundance presence of mosquitoes in these areas [1]." should be "...due to the abundance of mosquitoes in these areas [1]." |
Minor concerns are corrected as suggested |
Line 41 |
Line 43-44: please rephrase for better flow the following sentence: "Currently, 2.5 billion people worldwide are at risk for DENV infection, with a global annual infection of an estimated 390 million cases, including 500,000 hospitalizations and approximately 25,000 deaths." |
The sentence is rephased. |
Line 44 |
Line 65-66: "Probiotics are living microorganisms which both confer health benefits to the host when ingested in adequate amounts [11]." should be "Probiotics are living microorganisms that confer health benefits to the host when ingested in adequate amounts [11]." |
Minor concerns are corrected as suggested |
Line 81 |
Linne 120-121: Corrected Sentence: "PBMCs were isolated from whole blood using a BD Vacutainer CPT glass molecular diagnostics tube, according to the manufacturer’s protocol." |
Minor concerns are corrected as suggested |
Line 166 |
Line 123: "PBMCs were washed twice with phosphate-buffered saline (PBS) and aliquots of the PBMCs pellet were resuspended in Cellbanker2 (ZENOGEN, Japan) and cryopreserved at −80∘C Upon thawing, PBMCs were cultured..." should be "PBMCs were washed twice with phosphate-buffered saline (PBS), and aliquots of the PBMCs pellet were resuspended in Cellbanker2 (ZENOGEN, Japan) and cryopreserved at −80∘C. Upon thawing, PBMCs were cultured..." Line 152-153: "The Vero cells were seeded overnight at a density of 1×105cells/well in 24 well plates." should be "Vero cells were seeded overnight at a density of 1×105cells/well in 24-well plates." |
Minor concerns are corrected as suggested |
Line 168 Line 202 |
Line 153: "... monolayer confluent Vero cells were infected with ten-fold serial diluted DENV, CHIKV, and ZIKV for one hour at 37∘C fpr preadsorption." should be "... monolayer confluent Vero cells were infected with ten-fold serial diluted DENV, CHIKV, and ZIKV for one hour at 37∘C for preadsorption." |
Minor concerns are corrected as suggested |
Line 204 |
Line 171: "Plaques in each well was counted, and viral titer was determined in plaque-forming unit per microliter (PFU/mL) [36]." should be "Plaques in each well were counted, and viral titer was determined in plaque-forming unit per microliter (PFU/mL) [36]." |
Minor concerns are corrected as suggested |
Line 220 |
Line 287: "There was a statistically significant (non-parametric Kruskal-Wallis and Steel test; p<0.05) increase in the expression level of HLA-DR due to LC-Plasma treatment." should be "There was a statistically significant (non-parametric Kruskal-Wallis and Steel test; p<0.05) increase in the expression level of HLA-DR due to LC-Plasma treatment." |
Unfortunately, we did not spot any difference in this concern. |
NA |
Line 289: Sentence: "...the increased in CD86 levels in pDCs following LC-Plasma treatment..." Correction: "increased" should be "increase." |
Minor concerns are corrected as suggested |
Line 347 |
Line 387: “the increased of IFN-α” should be corrected to "the increase in IFN-α" "The increased of IFN-a was comparable to the effect seen with the representative TLR9 ligand CpG ODN 1585 [29]." should be "The increase in IFN-a was comparable to the effect seen with the representative TLR9 ligand CpG ODN 1585 [29]." |
Minor concerns are corrected as suggested |
Line 446 |
Line 394: "This suggested that LC-Plasma and CpG ODN 2216 might similarly stimulate PBMCs to produce the antiviral effects.[44]" should be "This suggested that LC-Plasma and CpG ODN 2216 might similarly stimulate PBMCs to produce the antiviral effects [44]." |
Minor concerns are corrected as suggested |
Line 463 |
Line 470: "This difference could be attributed to the potentially distinct IFNa subtypes produced by PBMCs upon LC-Plasma or CpG ODN treatment." should be "This difference could be attributed to the potentially distinct IFN-a subtypes produced by PBMCs upon LC-Plasma or CpG ODN treatment."
|
Minor concerns are corrected as suggested |
Line 540 |
Reviewer 3 Report
Comments and Suggestions for Authors
The manuscript (microorganisms-3287471), entitled “Heat-killed Lactococcus Lactis Strain Plasma Stimulates PBMCs to Elicit Interferon-Alpha Production and Promote Broad-Spectrum Antiviral Activity Against Dengue, Chikungunya, and Zika Viruses.
1. The study addresses an important topic in the field of antiviral research, focusing on the potential use of probiotics against emerging arboviruses like DENV, CHIKV, and ZIKV1
2. The experimental design is comprehensive, employing a variety of techniques to evaluate the antiviral effects of LC-Plasma, including PBMC stimulation, virus titration assays, and gene expression analysis.
3. While the immunomodulatory effects of LC-Plasma have been studied before, this research extends its application to major tropical arboviruses, which is a novel approach.
4. The study provides a detailed investigation into the mechanism of action, focusing on IFN-α production and subsequent ISG upregulation.
Based on my review of the manuscript, here are some key questions I would pose to the authors:
1. In vivo relevance: Have you considered conducting any animal studies to validate your in vitro findings? How might the results translate to an in vivo system?
2. Sample size: The study used PBMCs from 25 healthy volunteers. Can you justify this sample size? Have you performed any power calculations to ensure statistical significance?
3. Dosage optimization: Did you investigate different concentrations of LC-Plasma to determine the optimal dose for antiviral effects? If not, what was the rationale for choosing 10 μg/mL?
4. Comparative analysis: Have you considered comparing LC-Plasma with other known probiotics or antiviral compounds to better contextualize its efficacy?

Author Response
Comments |
Response and amendment |
Page number and Line number |
In vivo relevance: Have you considered conducting any animal studies to validate your in vitro findings? How might the results translate to an in vivo system? |
Thank you for raising this important point. We acknowledge the relevance of bridging our in vitro findings with in vivo implications. In our previous work, we already explored the anti-DENV effects of LC-Plasma using a murine model [1]. Building on that study, our current focus was to investigate the impact of LC-Plasma on human samples. Thus, we employed an in vitro model using humoral factors derived from PBMCs to investigate the broad spectrum antiviral effect of LC-Plasma in human samples. Additionally, we have reported on the preventive potential of orally administered LC-Plasma in mitigating dengue symptoms in a high-risk area in the Klang Valley, Malaysia [2]. These findings provide initial evidence for the systemic effects of LC-Plasma and support further exploration of its efficacy in human subjects. Further study will also have to be conducted to investigate the antiviral effect of LC-Plasma in mice model against CHIKV and ZIKV. Future study was added in the manuscript: Furthermore, future research directions could be expanded to clinical studies or in vivo models to test the efficacy of LC-Plasma in preventing arboviral infections. |
Page 13 Line 592 |
Sample size: The study used PBMCs from 25 healthy volunteers. Can you justify this sample size? Have you performed any power calculations to ensure statistical significance? |
In our previous study, we found that a small sample size (n=3) was sufficient to detect statistical differences when analyzing PBMCs stimulated with LC-Plasma in vitro [3]. The sample size of the current study was determined using parameters adopted from a previous study (Fujii et al., 2017) conducted by the same research team. In the previous study, peripheral blood mononuclear cells (PBMCs) were isolated from 200 participants following oral stimulation with LC-Plasma (control vs LC-Plasma). A statistical comparison between the two groups revealed that the level of IFN-α elicited by the A/H1N1 virus were higher in the LC-Plasma group compared to the control group. In this study, PBMCs were stimulated by direct inoculation of LC-Plasma into the PBMC culture. Ex vivo stimulation of PBMCs enables a higher degree of PBMC activation than oral consumption. Following the comments, the authors added effect size and type-error, and power (Effect size: 0.8-0.9, Type-1 error: 0.05 and power: 0.8-0.9). |
Page 3 Line 159-162 |
Dosage optimization: Did you investigate different concentrations of LC-Plasma to determine the optimal dose for antiviral effects? If not, what was the rationale for choosing 10 μg/mL? |
Thank you for your insightful questions. We chose the 10 µg/mL concentration of LC-Plasma based on its established safety profile, including its well-documented tolerance even in populations. Previous publications from our laboratory have consistently utilized this concentration for stimulating various in vitro cell types, such as PBMCs, mDCs, pDCs, and BM cells [1,3,5,6]. Employing the 10 µg/mL concentration facilitates meaningful comparisons across different studies, allowing for a more unbiased assessment of the physiological effects of LC-Plasma. References were added into the section 2.7 |
Page 4 line 235 |
Comparative analysis: Have you considered comparing LC-Plasma with other known probiotics or antiviral compounds to better contextualize its efficacy? |
Thank you for your valuable feedback. In response to your question, we would like to highlight that, in a comprehensive screening of 125 lactic acid bacteria, LC-Plasma stood out for its ability to significantly increase IFN-α production in murine bone marrow-derived dendritic cells (BM-DCs) [7]. Additionally, our previous studies have compared the immunomodulatory effects of LC-Plasma with other strains, including Lactococcus lactis JCM 1158, Lactobacillus acidophilus JCM 1132, Lactobacillus brevis JCM 1059, Lactobacillus casei JCM 1134, Lactobacillus gasseri JCM 1131, Lactobacillus helveticus JCM 1120, and Lactobacillus rhamnosus GG (ATCC 53103) [1]. These comparisons showed that LC-Plasma had the highest effect in enhancing IFN-α levels among the strains tested.Interestingly, LC-Plasma but not other LAB was observed microscopically to be internalized into plasmacytoid dendritic cells (pDCs) – a rare occurrence for bacteria of this size and shape, suggesting a potentially systemic effect [8]. As for the comparison with other antiviral compounds, we appreciate this suggestion. Currently, there are no FDA-approved antiviral drugs or compounds for DENV, CHIKV, and ZIKV. While the effects of LC-Plasma as a probiotic may not directly compare to those of traditional antiviral compounds, its high tolerance and safety profile make it a promising candidate. |
NA |